# Phenotypic and Genotypic screening of fifty-two rice (*Oryza sativa* L.) genotypes for desirable cultivars against blast disease

**Jeevan B.**[1]\*, **Rajashekara Hosahatti**[1], **Prasanna S. Koti**[2], **Vinaykumar Hargi Devappa**[3], **Umakanta Ngangkham**[4], **Pramesh Devanna**[5], **Manoj Kumar Yadav**[6], **Krishna Kant Mishra**[1], **Jay Prakash Aditya**[1], **Palanna Kaki Boraiah**[3], **Ahmed Gaber**[7], **Akbar Hossain**[8]\*

**1** ICAR-Vivekananda Parvatiya Krishi Anusandhan Sansthan, Almora, Uttarakhand, India, **2** The University of Trans-Disciplinary Health Sciences and Technology, Jarakabande Kaval, Bengaluru, Karnataka, India, **3** Project Coordinating Unit, ICAR-AICRP on Small Millets, UAS, GKVK, Bengaluru, Karnataka, India, **4** ICAR- Research Complex for North- Eastern Hill Region, Manipur centre, Imphal, Manipur, India, **5** Rice Pathology Laboratory, AICRIP, Gangavathi, University of Agricultural Sciences, Raichur, Karnataka, India, **6** ICAR-Indian Agricultural Research Institute, Regional Station, Karnal, Haryana, India, **7** Department of Biology, College of Science, Taif University, Taif, Saudi Arabia, **8** Department of Agronomy, Bangladesh Wheat and Maize Research Institute, Dinajpur, Bangladesh

\* jeevan.bscag@gmail.com (JB); akbar.hossain@bwmri.gov.bd (AH)

**Data Availability Statement:** All relevant data are within the paper.

## Abstract

*Magnaporthe oryzae*, the rice blast fungus, is one of the most dangerous rice pathogens, causing considerable crop losses around the world. In order to explore the rice blast-resistant sources, initially performed a large-scale screening of 277 rice accessions. In parallel with field evaluations, fifty-two rice accessions were genotyped for 25 major blast resistance genes utilizing functional/gene-based markers based on their reactivity against rice blast disease. According to the phenotypic examination, 29 (58%) and 22 (42%) entries were found to be highly resistant, 18 (36%) and 29 (57%) showed moderate resistance, and 05 (6%) and 01 (1%), respectively, were highly susceptible to leaf and neck blast. The genetic frequency of 25 major blast resistance genes ranged from 32 to 60%, with two genotypes having a maximum of 16 *R*-genes each. The 52 rice accessions were divided into two groups based on cluster and population structure analysis. The highly resistant and moderately resistant accessions are divided into different groups using the principal coordinate analysis. According to the analysis of molecular variance, the maximum diversity was found within the population, while the minimum diversity was found between the populations. Two markers (RM5647 and K39512), which correspond to the blast-resistant genes *Pi36* and *Pik*, respectively, showed a significant association to the neck blast disease, whereas three markers (Pi2-i, Pita3, and k2167), which correspond to the blast-resistant genes *Pi2*, *Pita/Pita2*, and *Pikm*, respectively, showed a significant association to the leaf blast disease. The associated R-genes might be utilized in rice breeding programmes through marker-assisted breeding, and the identified resistant rice accessions could be used as prospective donors for the production of new resistant varieties in India and around the world.

**Funding:** This research was funded by ICAR-Vivekananda Parvatiya Krishi Anusandhan Sansthan, Almora, Uttarakhand, India. This research was also partially funded by the Taif University Researchers for funding this research with Supporting Project number (TURSP-2020/39), Taif University, Taif, Saudi Arabia. The funders had no role in study design, data collection and analysis, decision to publish, or preparation of the manuscript.

**Competing interests:** The authors have declared that no competing interests exist

## Introduction

Rice blast disease, caused by filamentous fungus *Magnaporthe oryzae* (anamorph *Pyricularia oryzae*), remains a potential threat to global rice production [1, 2]. The blast pathogen can be found in all stages of plants growth and development, causing damage to leaves (leaf blast), nodes (nodal blast), and panicles (neck blast), as well as decreasing grain yield by up to 90% in favourable environmental conditions [3–5].

The *M. oryzae* has been documented all over the world and can infect more than 50 host species in the family Poaceae, including rice, wheat, pearl millet, foxtail millet, and finger millet [6–8]. Across most of the world's rice-growing regions, including India, blast disease epidemics have occurred [9, 10]. Between 1980 and 1987, India experienced several deadly blast disease epidemics in Himachal Pradesh, Tamil Nadu, Andhra Pradesh, and Haryana [11, 12].

Chemical fungicides have been useful in controlling the disease, but they are expensive [13, 14], ineffective when disease pressure is high [15], and may contribute to pathogen resistance [16]. As a result, the most cost-effective and environmentally acceptable strategy for controlling rice blast disease is to leverage host resistance (*R* genes). Around 118 *R* genes have been discovered so far, with 35 of them being successfully cloned and characterized for leaf blast resistance [17, 18]. However, the cloned *R* genes that possess broad-spectrum resistance to leaf blast, have not been tested for neck blast disease [19]. Even though neck blast is the most devastating stage of the disease, there is relatively little information on the genetic processes that underpin neck blast resistance. Nevertheless, 14 QTLs [18] and a few *R* genes have been found for neck blast resistance, including *Pi25(t)* [20], *Pb1* [21], *Pi64* [22], *Pi-jnw1* [23], and *Pi68(t)* [24]. A large majority of the cloned blast *R* genes share nucleotide-binding site (NBS) and leucine-rich repeat (LRR) domains in their protein sequences, except for a few (*Pid2*, *pi21*, and *Ptr*) [17, 25, 26]. According to gene-for-gene theory, these *R* genes are race-specific and related to the hypersensitive response (HR) [27]. The *M. oryzae*'s genome contains numerous repetitive DNA and retro-transposons [28], which might cause mutations in genes that mediate the pathogen's virulence and host range [29–31], allowing the fungus to develop new deadly races. The emergence of these races results in a change in pathogenicity, posing a threat to existing blast-resistant rice cultivars [32].

By permitting the integration of the desired gene(s) in early breeding generations, marker-assisted selection (MAS) has emerged as a potent method that has advanced the rice breeding effort for blast disease resistance [33]. Many rice cultivars have been improved *via* MAS by pyramiding targeted *R* genes, resulting in the rapid release of rice varieties with durable resistance against blast disease [34]. In recent years, molecular markers have been utilized to capitalize on natural variety and pinpoint the gene of interest influencing essential features in different germplasm [35].

There is indeed a lot of genetic variation in the Indian rice germplasm collection [12, 36]. Many of these rice varieties have been reported to have resistance to biotic and abiotic stresses, including blast disease [37–39]. However, the distribution of *R* genes in Indian rice cultivars that confer long-term resistance to leaf and neck blast has not been adequately explored. As a result, it's critical to comprehend *R* gene information in rice germplasm as well as the resistant spectrum of relevant *R* genes against prevailing pathogen races to use the most successful ones in the rice breeding programme to combat blast disease. The present study was carried out to explore the genetic association of 25 mapped resistance genes in 52 rice accessions, including released varieties, advanced breeding materials, and traditional rice varieties using linked/functional markers. The main goal of this study was to find an association between the leaf and neck blast *R* genes, which impart blast resistance to these lines, and novel blast resistance donor sources (*R* genes/alleles).

**Table 1. List of 52 rice accession used in this study.**

| Planting materials | Genotypes |
|---|---|
| Released varieties | VL *Dhan* 158, VL *Dhan* 68, VL *Dhan* 221 and VL *Dhan* 206 |
| Advanced breeding materials | VL 8083, VL 8214, VL 8394, VL 8549, VL 8654, VL 20231, VL 20279, VL 20287, VL 20298, VL 20299, VL 20302, VL 20289, VL 31430, VL 31451, VL 31598, VL 31615, VL 31616, VL 31619, VL 31674, VL 31679, VL 31694, VL 31716, VL 31743, VL 31802, VL 31817, VL 31851, VL 31870, VL 31916, VL 31997, VL 32092, VL 32131, VL 32132, VL 32168, A-57, BL-122, BL-245, GSR-102, GSR-106, GSR-124, GSR-125, GSR-132, GSR-142, VOHP-3102 and VL 32197 |
| Traditional rice varieties | VLK 39 and Someshwar |
| Susceptible checks | PB-1 and Bala |

## Materials and methods

### Plant materials used in the current research

A total of 50 rice accessions were collected based on documented rice blast resistant information from the Rice Genetics laboratory, Crop Improvement Division, ICAR- Vivekananda Parvatiya Krishi Anusandhan Sansthan (VPKAS), Almora, Uttarakhand, India (Tables 1 & 2). The test material includes released varieties (04), advanced breeding materials (44), and traditional rice varieties (02). In addition, two genotypes, PB-1 and Bala, were chosen as leaf and neck blast susceptible controls, respectively (Tables 1 and 2).

**Table 2. List of rice genotypes along with their pedigree.**

| Sl. No. | Entry name | Pedigree | Sl. No. | Entry name | Pedigree |
|---|---|---|---|---|---|
| 1 | VL 8083 | VL 6394/VL 6446 | 27 | VL 31817 | Vivek *Dhan* 82/BL122 |
| 2 | VL 8214 | VL Dhan 81/VR539-2 | 28 | VL 31851 | VL 30424/IR78 |
| 3 | VL 8394 | VL6394/VL6446 | 29 | VL 31870 | BL 122/IR 785–36 |
| 4 | VL 8549 | VL 3861/VL 6394 | 30 | VL 31916 | VL *Dhan* 85/BL 245 |
| 5 | VL 8654 | RCPL 1-45/Vivek Dhan 154 | 31 | VL 31997 | Vivek *Dhan* 62/MAS-52 |
| 6 | VL *Dhan* 158 | RCPL 1-45/VL 3861 | 32 | VL 32092 | VL *Dhan* 85/VOHP 3102 |
| 7 | VL 20231 | VL *Dhan* 81/Vandana | 33 | VL 32131 | VL 10689/UPRI2005-15 |
| 8 | VL 20279 | VL 20240/Sawdhan | 34 | VL 32132 | VL 10689/UPRI2005-15 |
| 9 | VL 20287 | VHC 1462/VL 10499 | 35 | VL 32168 | VL *Dhan* 65/VL30919 |
| 10 | VL 20298 | Annada/C101-A51 | 36 | A-57 | - |
| 11 | VL 20299 | Annada/C101-A51 | 37 | BL-122 | - |
| 12 | VL 20302 | VL *Dhan* 221/ VL 30927 | 38 | BL-245 | - |
| 13 | VL 20289 | VHC 1462/VL 10499 | 39 | VL *Dhan* 221 | IR 2053-521-1-1-1/Ch 1039 |
| 14 | VL 31430 | Pant *Dhan* 6/VL 3288 | 40 | VLK 39 | China 1039/IR580-19-2-3-1 |
| 15 | VL 31451 | IR 72979/PSB RC 2 (IR 32809-26-3-3) | 41 | GSR-102 | - |
| 16 | VL 31598 | VL 3861/IR57257-34-1-2-1 | 42 | GSR-106 | - |
| 17 | VL *Dhan* 68 | VL 3861/SR 1818BF-4B-1-2-1-2 | 43 | GSR-124 | - |
| 18 | VL 31615 | VL 3861/SR 1818BF-4B-1-2-1-2 | 44 | GSR-125 | - |
| 19 | VL 31616 | VL 3861/SR 1818BF-4-B1-2-1-2 | 45 | GSR-132 | - |
| 20 | VL 31619 | VL 3861/SR 1818BF-4-B1-2-1-2 | 46 | GSR-142 | - |
| 21 | VL 31674 | C101-A51/*O. minuta* | 47 | VOHP-3102 | Local collection |
| 22 | VL 31679 | *O. minuta*/Vivek *Dhan* 82 | 48 | VL *Dhan* 206 | Pure line selection from Bamni (local variety) |
| 23 | VL 31694 | Vivek *Dhan* 82/IR57257-34 | 49 | VL 32197 | VL *Dhan* 81/Vandana |
| 24 | VL 31716 | *O. minuta*/IR57257-34 | 50 | Someshwar | Local collection |
| 25 | VL 31743 | VL 30424/IR32809 | 51 | Bala | N 22/T(N)1 |
| 26 | VL 31802 | VL 66/VL30424 | 52 | PB-1 | Pusa 167/Karnal Local |

## Phenotyping of rice germplasm lines for blast disease resistance

A set of 50 rice hill germplasm collections were evaluated under the natural conditions at the rice blast hotspot area, ICAR-VPKAS, experimental farm, Hawalbagh (29o56'N, 79o40'E, and 1250m MSL), Almora, for their reactivity against leaf and neck blast. The evaluations were carried out in three replications over three years, from 2018 to 2020, during the rainy (Kharif) seasons. Sowings were done in two sets, one for leaf blast evaluations and the other for neck blast evaluations. Each rice entry (30 plants/test entry) was raised in 50 cm long rows on nursery beds with a 10 cm row spacing in a uniform blast nursery for leaf blast (UBN). One line of PB-1 (susceptible check) was sown after every 5 entries of test accessions, as well as along the boundaries, to ensure adequate disease transmission. From 25 days after sowing until the susceptibility check showed 85% of the blast disease symptom, the disease spectrum of all the test entries was recorded. A 0–9 scale devised by IRRI, Philippines [39], was used to visually record the disease reaction on each test entry.

Similarly, the other set was also tested for neck blast disease, but Bala was used as a susceptible control. The severity of the disease was graded on a 0–9 scale (IRRI, 2002), with 0 = no lesion or one or two tiny lesions on the panicles; 1 = symptom on several pedicels or secondary branches; 3 = lesions on a few primary branches or the middle part of panicle axis; 5 = moderate infection with lesions covering half of the node or the uppermost internode or the lower part of panicle axis; 7 = heavy infection, lesions abundant on the panicle base or uppermost internode or panicle axis near the base with more than 30% of filled grains; 9 = very heavy infection, around the panicle base or uppermost internode or the panicle axis near the base with less than 30% of filled grains. At physiological maturity, the disease reaction was recorded, and the affected plants were evaluated on a disease scale, Highly resistant (HR) (0–3 score), moderately resistant (MR) (4–5), and susceptible (S) (6–9) were assigned to the test entries, respectively. Whenever differences in the disease spectrum were recorded, the higher disease was taken into account.

## DNA isolation and genotyping

Genomic DNA was extracted from the young leaves of 50 rice germplasm lines and two susceptible controls using the CTAB technique [40]. The quality and quantity of isolated genomic DNA were determined using a Thermo Fisher Scientific NanoDrop™ 1000 Spectrophotometer. After that, the isolated DNA samples were diluted to a concentration of 25 ng/μl in nuclease-free water for PCR amplification. Molecular profiling of 52 rice lines for the presence of major blast resistance genes was carried out using 25 linked or functional molecular markers. The detailed information on blast resistance genes and their corresponding primer pairs used in this investigation is listed in Table 2. About 25 ng of template DNA, 10 pmol of each forward and reverse primers, 25 mM $MgCl_2$, 2 mM of each dNTPs, 1X Taq buffer, 1U Taq DNA polymerase, and nuclease-free water were used in the PCR amplification. The PCR conditions were set as follows: initial denaturation at 94˚C for 5 minutes was followed by 35 cycles of denaturation for 40 seconds at 94˚C, primer annealing for 40 seconds at varied temperatures (Table 3), and extension for 2 minutes at 72˚C were performed, followed by a final 10-minute extension at 72˚C. To double-check the results, PCR amplification was done twice for each marker. The amplified PCR products were resolved in ethidium bromide-stained 3% agarose gels and the scoring were done for the PCR analysis as presence (1) or absence (0).

## Allele scoring and genetic diversity analysis

The presence or absence of an allele was indicated as 1 and 0, respectively, in the amplified PCR products of 25 markers, which were scored as a binary matrix. Using a binary data matrix

**Table 3. Details of markers used for molecular screening of blast resistance genes in 52 rice accessions.**

| Genes | Markers | Forward (5' - 3') | Reverse (5' - 3') | Type of Marker[*] | Annealing Temperature (˚C) | References |
|---|---|---|---|---|---|---|
| **Pit** | tk59-1 | ATGATAACCTCATCCTCAATAAGT | GTTGGAGCTACGGTTGTTCAG | FM | 54 | [48] |
| **Pid1(t)** | RM262 | CATTCCGTCTCGGCTCAACT | CAGAGCAAGGTGGCTTGC | LM | 55 | [63] |
| **Pish** | RM6648 | GATCGATCATGGCCAGAGAG | ACAGCAGGTTGATGAGGACC | LM | 55 | [34] |
| **Pb1** | RM26998 | ACGCACGCACATCCTCTTCC | CGGTTCTCCATCTGAAATCCCTAGC | LM | 55 | [21] |
| **Pi33** | RM72 | CCGGCGATAAAACAATGAG | GCATCGGTCCTAACTAAGGG | LM | 55 | [64] |
| **Pikhahe-1 (t)** | RM17496 | TAAACGGTGTGCAGCTTCTG | TATTATGGGCGGTCGCTAAC | LM | 54 | [65] |
| **pi21** | pi21-79-3 | GATCCTCATCGTCGACGTCTGGC | AGGGTACGGCACCAGCTTG | InDel | 55 | [27] |
| **Pi56** | CRG4-2 | CCTGTCAGTCTTTCCGAGAG | GAATCCGGTAGCTCAAGGTG | Gene-specific | 55 | [66] |
| **Pi65** | SNP_3 | TGCCACCAGCCATCTTCAACAT | ACCACATCACTCATCGCCATCC | InDel | 54 | [71] |
| **Pi36** | RM5647 | ACTCCGACTGCAGTTTTTGC | AACTTGGTCGTGGACAGTGC | LM | 55 | [72] |
| **Pi49** | RM6094 | TGCTTGATCTGTGTTCGTCC | TAGCAGCACCAGCATGAAAG | LM | 55 | [67] |
| **Pi48** | RM5364 | GTATTACGCTCGATAGCGGC | GTATCCTTTCTCGCAATCGC | LM | 55 | [68] |
| **Pib** | Pb28 | GACTCGGTCGACCAATTCGCC | ATCAGGCCAGGCCAGATTTG | SNP | 60 | [48] |
| **Piz** | Z56592 | GGACCCGCGTTTTCCACGTGTAA | AGGAATCTATTGCTAAGCATGAC | SNP | 60 | [48] |
| **Piz-t** | Zt56591 | TTGCTGAGCCATTGTTAAACA | ATCTCTTCATATATATGAAGGCCAC | SNP | 60 | [48] |
| **Pik** | K39512 | GCCACATCAATGGCTACAACGTT | CCAGAATTTACAGGCTCTGG | SNP | 60 | [48] |
| **Pik-p** | K3957 | ATAGTTGAATGTATGGAATGGAAT | CTGCGCCAAGCAATAAAGTC | SNP | 60 | [48] |
| **Pik-h** | Candidate gene marker | CATGAGTTCCATTTACTATTCCTC | ACATTGGTAGTAGTGCAATGTCA | Gene-based marker | 55 | [69] |
| **Pi9** | Pi9-i | GCTGTGCTCCAAATGAGGAT | GCGATCTCACATCCTTTGCT | FNP | 54 | [52] |
| **Pi2** | Pi2-i | CAGCGATGGTATGAGCACAA | CGTTCCTATACTGCCACATCG | FNP | 52 | [52] |
| **Pita/Pita2** | Pita3 | AGTCGTGCGATGCGAGGACAGAAAC | GCATTCTCCAACCCTTTTGCATGCAT | SNP | 59 | [48] |
| **Pi1** | RM1233 | GTGTAAATCATGGGCACGTG | AGATTGGCTCCTGAAGAAGG | SSR | 55 | [40] |
| **Pi5** | 40N23R | TGTGAGGCAACAATGCCTATTGCG | CTATGAGTTCACTATGTGGAGGCT | InDel | 55 | [40] |
| **Pikm** | k2167 | CGTGCTGTCGCCTGAATCTG | CACGAACAAGAGTGTGTCGG | InDel | 55 | [40] |
| **Pi25** | CAP1 | TGAAATGGGTGAAAGATGAG | GCCACATCATAATTCCTTGA | CAPS | 55 | [70] |

[*] FM, functional marker; LM, linked marker; InDel, insertion-deletion marker; FNP, functional nucleotide polymorphism; SNP, single nucleotide polymorphism; CAPS, Cleaved Amplified Polymorphism Sequences

of 25 markers, the genetic distance and similarity coefficients for 52 rice accessions were calculated. Using the Cervus 3.0 programme (Field Genetics Ltd., London, England) and POP-GENE 32 software, different parameters such as the number of different alleles per locus (Na), number of effective alleles per locus (Ne), Shannon's Information Index (I), and Expected Heterozygosity ($H_E$) for each marker were calculated [41]. Subsequently, a heatmap of all the rice accessions was constructed using the pheatmap package with complete linkage clustering method and euclidean distance measure by R version 4.0.3 statistical software for the presence or absence of 25 markers for both leaf and neck blast.

## Association analysis

To study the genetic relationship between blast resistance genes and the disease spectrum, we used TASSEL version 5.0 software with a general linear model (GLM) function [42]. Only the *P*-value was seen in 5% of the permutations for the most significant polymorphism in a region when the GLM model of TASSEL (v 5.0) software was performed with 1000 permutations of data. Using genotypic data collected with 25 molecular markers and pheatmap-based clustering with complete linkage clustering method and Euclidean distance measure, the genetic

distance between the 52 rice accessions was estimated using R version 4.0.3 statistical programme.

## Population structure analysis

Based on genotyping data from 25 markers, the STRUCTURE software v 2.3.4 [43] was used to evaluate the population structure of 52 rice accessions. Using the admixture and correlated allele frequencies model, each subpopulation (K) was estimated at different *K* values ranging from one to ten, with five runs per *K* value. A total of 200000 burn-in periods and 200,000 Markov chain Monte Carlo (MCMC) iterations were used in the STRUCTURE runs. Using the STRUCTURE HARVESTER software, the highest delta K (*ΔK*) value was estimated to determine the most likely *K*-value [44]. The pairwise fixation index (FST) was calculated using principal coordinate analysis (PCoA) based on a binary data matrix of 25 markers, and analysis of molecular variance (AMOVA) was performed using the GenAlEx version 6.502 software [45].

## Results

### Phenotyping of hill germplasm lines

Initially, the responsiveness of 277 rice accessions to rice blast disease was assessed. From these 277 accessions, we chose 52 genotypes based on their reaction to rice blast disease. i.e., resistant, moderately resistant, and susceptible (Tables 1 & 2).

Of 52 rice genotypes, 29 (58%) and 22 (42%) rice genotypes were found to be highly resistant, 18 (36%) and 29 (57%) were moderately resistant, while 05 (6%) and 01 (1%) were highly susceptible to leaf and neck blast, respectively. Incidentally, sixteen genotypes showed high resistance to both leaf and neck blasts (Fig 1).

### Genetic diversity of blast-resistant R genes

The present study used a set of twenty-five markers (functional/linked markers) that corresponded to the twenty-five *R* genes (Table 3). The gene frequency of the twenty-five blast *R* genes ranged from 32 to 60%, with the number of positive *R*-gene alleles ranging from 0 to 100%. Using a tk59-1 marker to visualize a 733 bp amplicon, the rice blast *R*-gene *Pit* was discovered in 17 rice genotypes. *Pish* on chromosome 1 was amplified with marker RM6648, resulting in a 207-bp band that was detected in 23 genotypes.

A 137-bp amplicon corresponding to the RM26998 marker was used to find the *Pb1* gene on chromosome 11 in 19 genotypes. In 29 rice germplasm lines, the marker RM17496 was able to amplify the *Pikhahe-1(t)* gene with a fragment size of 84 bp. For the recessive blast-resistant gene *pi21*, only four genotypes were determined to be positive. The existence of the blast resistance gene *Pi56* on chromosome 9 was detected using the gene-specific marker CRG4-2, which was found in 23 genotypes. Using the linked marker RM5647, the blast resistance genes *Pi36* (chromosome 8) were discovered in 12 genotypes. The *Pi49* gene, which is located on chromosome 11, was found in 12 genotypes after 182 bp were seen with the RM6094 marker. Using the RM5364 primer, the *Pi48* gene was discovered in five genotypes.

The *R* genes, *Piz*, and *Piz-t* on chromosome 6 were amplified using SNP markers Z56592 and Zt56591, which revealed their presence in nine and twelve entries, respectively. Visualization of 112 bp, 148 bp, and 1500 bp amplicons corresponding to the K39512, K3957, and a gene-based marker, respectively, revealed the *Pik*, *Pik-p*, and *Pik-h* genes on chromosome 11. The genes *Pik*, *Pik-p*, and *Pik-h* were found in 46, 47, and 17 accessions, respectively. The *Pi2* gene was discovered using the Pi2-i primer in twenty-one entries, resulting in positive bands. The major blast resistance gene *Pita/Pita2*, which was scored by visualization of an 861 bp

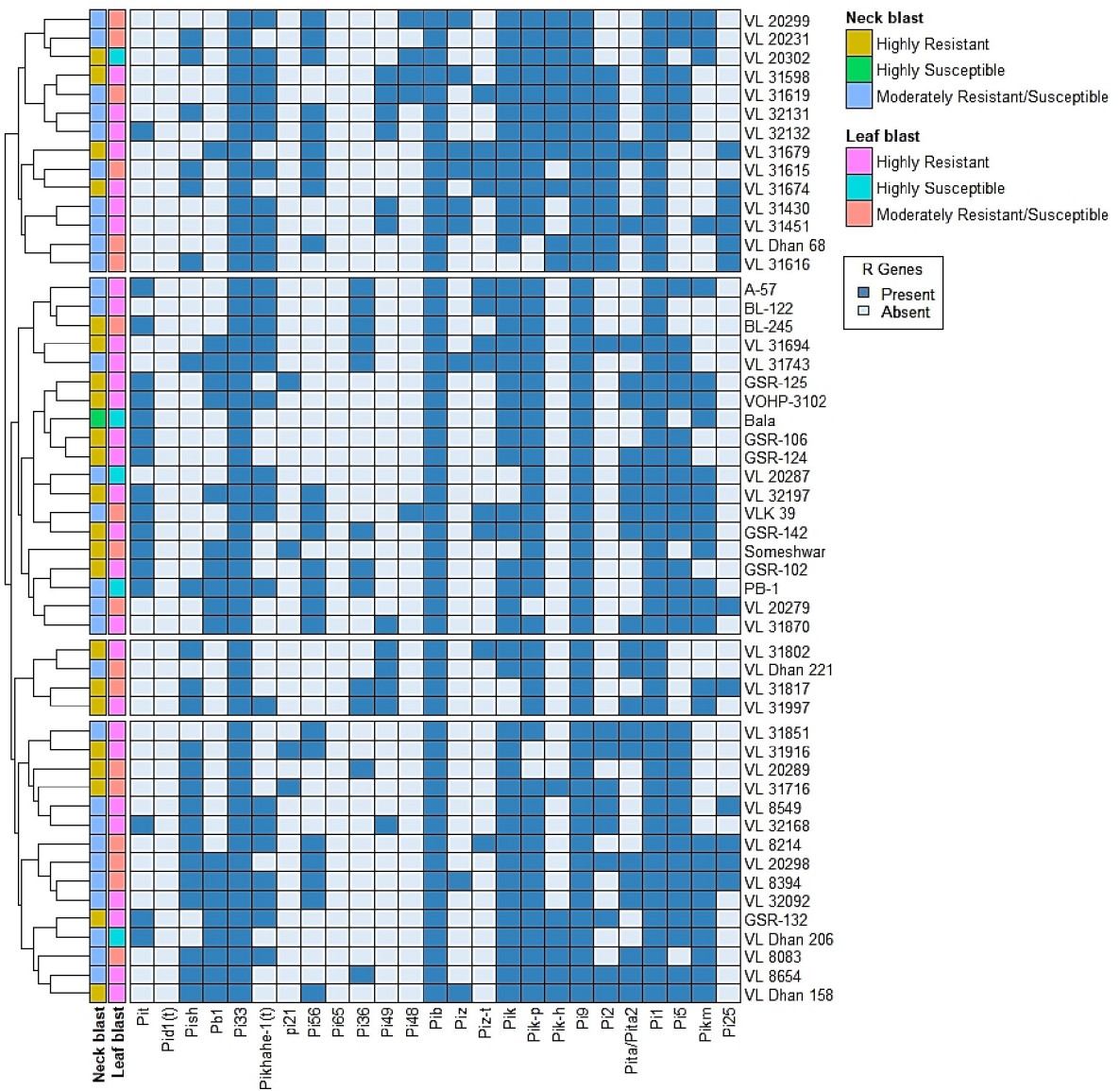

**Fig 1. A clustered analysis based on the 25 molecular markers and Heatmap representing the summary of phenotypic and genotypic data of 52 rice genotypes analyzed in this study.**

amplicon utilizing the Pita3 marker, was found in 22 genotypes. The *Pi5* gene was discovered in 35 genotypes, which was confirmed using the marker 40N23R. The *Pikm* gene was found in twenty-seven genotypes after PCR amplification. *Pi25* was found in twelve genotypes using the CAP1 primer, which produced a 406-bp amplicon. The R genes *Pi33*, *Pib*, *Pi9*, and *Pi1* were detected in all genotypes; however, the *Pid1(t)* and *Pi65* genes were not discovered in any of the fifty-two genotypes examined in this study. The phenotypic and genotypic data of the 52 rice accessions studied in this investigation are summarized in Fig 1.

## Cluster analysis

R-software was used to do the cluster analysis, which separated the 52 rice accessions into two primary clusters. Cluster I had 14 genotypes, seven and four of which were found to be highly

resistant to leaf and neck blast, respectively, and three of which, VL31598, VL31679, and VL31674, were shown to be resistant to both leaf and neck blast. Cluster II was divided into three subgroups, the first of which contained a large number of genotypes (19), including nine genotypes resistant to both leaf and neck blast. On the other hand, two susceptible checks (Bala and PB-1) were also clustered together. Subgroup II is made up of four genotypes: VL 31802, VL 31817, VL 31997, and VL *Dhan* 221. Except for VL *Dhan* 221, all three genotypes are resistant to neck blast. Subgroup III has fifteen genotypes, eight and five genotypes showed high resistance to leaf and neck blast, respectively, and three of which were common for both leaf and neck blast resistance, including VL *Dhan* 158, GSR-132, and VL31916. Except for VL *Dhan* 206, the majority of the genotypes exhibited moderate resistance to either leaf or neck blast (Fig 1).

The genotypic data from the 25 markers was used to calculate genetic diversity measures including the number of distinct alleles per locus (Na), the number of effective alleles per locus (Ne), Shannon's Information Index (I), and Expected Heterozygosity (HE). A total of 44 alleles were generated from 25 loci or markers (Table 4). The average number of alleles per locus (Na) was 1.76, with a range of 1 to 2. The number of effective alleles per locus (Ne) ranged from 1 to 1.99, with an average of 1.49. Shannon's Information Index (I) ranged from 0 to 0.692 (Pikm), with an average of 0.42. The Expected Heterozygosity (HE) ranged from 0 (Pid1 (t), Pi33, Pi65, Pib, Pi9(Pi9-i), and Pi1) to 0.499 (Pikm) with an average of 0.285.

**Table 4. Analysis of the number of alleles, Shannon's Information Index, observed and expected Heterozygosity.**

| Locus | Na | Ne | I | He |
|---|---|---|---|---|
| Pit | 2.000 | 1.786 | 0.632 | 0.440 |
| Pid1(t) | 1.000 | 1.000 | 0.000 | 0.000 |
| Pish | 2.000 | 1.974 | 0.686 | 0.493 |
| Pb1 | 2.000 | 1.865 | 0.656 | 0.464 |
| Pi33 | 1.000 | 1.000 | 0.000 | 0.000 |
| Pikhahe-1(t) | 2.000 | 1.974 | 0.686 | 0.493 |
| pi21 | 2.000 | 1.166 | 0.271 | 0.142 |
| Pi56 | 2.000 | 1.974 | 0.686 | 0.493 |
| Pi65 | 1.000 | 1.000 | 0.000 | 0.000 |
| Pi36 | 2.000 | 1.550 | 0.540 | 0.355 |
| Pi49 | 2.000 | 1.550 | 0.540 | 0.355 |
| Pi48 | 2.000 | 1.210 | 0.317 | 0.174 |
| Pib | 1.000 | 1.000 | 0.000 | 0.000 |
| Piz | 2.000 | 1.401 | 0.461 | 0.286 |
| Piz-t | 2.000 | 1.550 | 0.540 | 0.355 |
| Pik | 2.000 | 1.257 | 0.358 | 0.204 |
| Pik-p | 2.000 | 1.210 | 0.317 | 0.174 |
| Pik-h | 2.000 | 1.786 | 0.632 | 0.440 |
| Pi9 (Pi9-i) | 1.000 | 1.000 | 0.000 | 0.000 |
| Pi2 (Pi2-i) | 2.000 | 1.929 | 0.675 | 0.482 |
| Pita (Pita3) | 2.000 | 1.954 | 0.681 | 0.488 |
| Pi1 | 1.000 | 1.000 | 0.000 | 0.000 |
| Pi5 | 2.000 | 1.786 | 0.632 | 0.440 |
| Pikm | 2.000 | 1.997 | 0.692 | 0.499 |
| Pi25 | 2.000 | 1.550 | 0.540 | 0.355 |

## Association analysis

The genetic association of markers with leaf and neck blast disease was examined using the general linear model (GLM) function to see if there was any evidence of a significant link between gene-specific markers and the disease reaction. Only two markers (RM5647 and K39512), which correspond to the blast-resistant genes *Pi36* and *Pik*, respectively, showed a significant association with the neck blast disease, while only three markers (Pi2-i, Pita3, and k2167), which correspond to the blast-resistant genes *Pi2*, *Pita/Pita2*, and *Pikm*, respectively, showed a significant association with the leaf blast disease (Table 5). For leaf blast, the associated markers showed a phenotypic variance of 7.2% to 12.2%. The marker k2167, which is linked to the *Pikm* gene, was shown to have the maximum phenotypic variance. The markers K39512 and RM5647, corresponding to the blast-resistant genes *Pik* and *Pi36*, respectively, showed a phenotypic variance of 4.7 and 5.2% for neck blast. The remaining twenty markers, on the other hand, showed no significant association with blast disease ($p \leq 0.1$).

## Population structure analysis

Using STRUCTURE software, all 52 rice genotypes were examined for population structure estimation for leaf and neck blast disease based on 25 markers. The Adhoc Measure *K* peak

**Table 5. Genetic association of blast resistant genes with rice neck and leaf blast disease in 52 genotypes.**

| Marker | Neck blast | | Leaf blast | |
|---|---|---|---|---|
| | *P* value | marker_R$^2$ | *P* value | marker_R$^2$ |
| Pit | 0.31011 | 0.0206 | 0.94482 | 9.68E-05 |
| Pid1(t) | NaN | 0 | NaN | 0 |
| Pish | 0.97709 | 1.67E-05 | 0.76334 | 0.00183 |
| Pb1 | 0.7654 | 0.0018 | 0.97428 | 2.10E-05 |
| Pi33 | NaN | 0 | NaN | 0 |
| Pikhahe-1(t) | 0.26063 | 0.02524 | 0.77591 | 0.00164 |
| pi21 | 0.23418 | 0.02818 | 0.98122 | 1.12E-05 |
| Pi56 | 0.85031 | 7.19E-04 | 0.40692 | 0.0138 |
| Pi65 | NaN | 0 | NaN | 0 |
| Pi36 | 0.1001 | **0.05253**[*] | 0.46899 | 0.01054 |
| Pi49 | 0.74584 | 0.00212 | 0.16333 | 0.03849 |
| Pi48 | 0.66804 | 0.00371 | 0.18689 | 0.03458 |
| Pib | NaN | 0 | NaN | 0 |
| Piz | 0.62943 | 0.00469 | 0.31383 | 0.02028 |
| Piz-t | 0.93256 | 1.45E-04 | 0.16943 | 0.03742 |
| Pik | 0.10002 | **0.0479**[*] | 0.1949 | 0.03337 |
| Pik-p | 0.84113 | 8.11E-04 | 0.46584 | 0.01069 |
| Pik-h | 0.78264 | 0.00154 | 0.90339 | 2.98E-04 |
| Pi9 (Pi9-i) | NaN | 0 | NaN | 0 |
| Pi2 (Pi2-i) | 0.43981 | 0.01198 | 0.01374 | **0.1154**[**] |
| Pita (Pita3) | 0.62468 | 0.00482 | 0.05363 | **0.07247**[**] |
| Pi1 | NaN | 0 | NaN | 0 |
| Pi5 | 0.77264 | 0.00168 | 0.33685 | 0.01846 |
| Pikm | 0.1383 | 0.04341 | 0.01114 | **0.12202**[**] |
| Pi25 | 0.64389 | 0.00431 | 0.6185 | 0.005 |

[*] & [**] Significant at *P* value <0.1 and <0.05 respectively

plateau was discovered to be $K = 2$ (Fig 2), indicating that the complete 52 rice genotypes were divided into two subgroups (SG1 and SG2).

All populations were divided into two major subgroups with eight admixture levels based on an ancestry threshold of >60% (Table 6). SG1 was made up of the most genotypes identified to be highly resistant to neck blast. The majority of genotypes identified to be highly resistant to leaf blast, on the other hand, were concentrated in SG2. Genotypes with moderate resistance to both leaf and neck blast were clustered together in SG2, while genotypes with high susceptibility to both leaf and neck blast were grouped together in SG1.

PCoA analysis has been carried out to establish the genetic relationship among the rice genotypes. PCoA analysis revealed that the first two axes explained 17.18% and 12.29% of the total variance (Table 7 and Fig 3). In PCoA, leaf blast-resistant genotypes were largely distributed among 1st and 2nd quadrants; on the other hand, most genotypes showed neck blast-resistant were concentrated in the 2nd quadrant. The genotypes found moderately resistant to both leaf and neck blast resistance were mostly distributed among the 1st, 3rd, and 4th quadrants, whereas susceptible genotypes were concentrated in the 2nd quadrant.

## AMOVA analysis

The genetic variations within and between the populations were assessed using AMOVA analysis. The leaf blast score was used to separate 52 rice genotypes into three populations: 29 (HR), 18 (MR), and 05 (S). Similarly, based on neck blast score, 22(HR), 29(MR), and 01(S) were separated. Furthermore, the maximum variance (91%) and (89%) was found within the population, while the least (9%) and (11%), respectively, were found between the populations for leaf and neck blast scores (Fig 4).

## Discussion

Rice genetic diversity has been reduced as a result of large-scale cultivation of high-yielding rice varieties, which have replaced landraces and traditional cultivars, limiting varietal

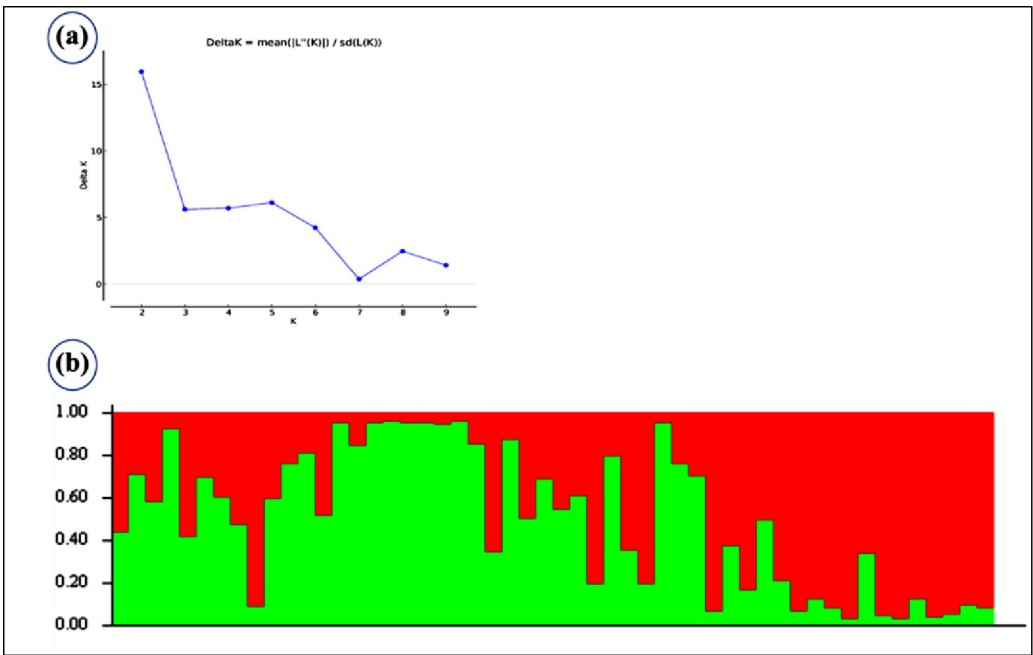

**Fig 2.** Population structure analysis of 52 rice genotypes (a) The maximum of ad hoc measure *ΔK* was observed to be *K* = 3 (b) Estimated population structure graph separated the whole population into two subgroups.

**Table 6. Population structure group of 52 genotypes based on inferred ancestry values.**

| Genotypes | Inferred Ancestry | | Structure group |
|---|---|---|---|
| | Q1 | Q2 | |
| VL 8083 | 0.560 | 0.44 | AD |
| VL 8214 | 0.291 | 0.709 | SG2 |
| VL 8394 | 0.419 | 0.581 | AD |
| VL 8549 | 0.076 | 0.924 | SG2 |
| VL 8654 | 0.582 | 0.418 | AD |
| VL *Dhan* 158 | 0.302 | 0.698 | SG2 |
| VL 20231 | 0.398 | 0.602 | SG2 |
| VL 20279 | 0.527 | 0.473 | AD |
| VL 20287 | 0.913 | 0.087 | SG1 |
| VL 20298 | 0.403 | 0.601 | SG2 |
| VL 20299 | 0.237 | 0.763 | SG2 |
| VL 20302 | 0.186 | 0.814 | SG2 |
| VL 20289 | 0.484 | 0.516 | AD |
| VL 31430 | 0.047 | 0.953 | SG2 |
| VL 31451 | 0.151 | 0.849 | SG2 |
| VL 31598 | 0.045 | 0.955 | SG2 |
| VL *Dhan* 68 | 0.039 | 0.961 | SG2 |
| VL 31615 | 0.048 | 0.952 | SG2 |
| VL 31616 | 0.05 | 0.95 | SG2 |
| VL 31619 | 0.055 | 0.945 | SG2 |
| VL 31674 | 0.04 | 0.96 | SG2 |
| VL 31679 | 0.145 | 0.855 | SG2 |
| VL 31694 | 0.657 | 0.343 | SG1 |
| VL 31716 | 0.123 | 0.877 | SG2 |
| VL 31743 | 0.495 | 0.505 | AD |
| VL 31802 | 0.313 | 0.687 | SG2 |
| VL 31817 | 0.453 | 0.547 | AD |
| VL 31851 | 0.388 | 0.612 | SG2 |
| VL 31870 | 0.806 | 0.194 | SG1 |
| VL 31916 | 0.201 | 0.799 | SG2 |
| VL 31997 | 0.647 | 0.353 | SG1 |
| VL 32092 | 0.807 | 0.193 | SG1 |
| VL 32131 | 0.044 | 0.956 | SG2 |
| VL 32132 | 0.24 | 0.76 | SG2 |
| VL 32168 | 0.296 | 0.704 | SG2 |
| A-57 | 0.933 | 0.067 | SG1 |
| BL-122 | 0.626 | 0.374 | SG1 |
| BL-245 | 0.832 | 0.168 | SG1 |
| VL *Dhan* 221 | 0.503 | 0.497 | AD |
| VLK 39 | 0.79 | 0.21 | SG1 |
| GSR-102 | 0.933 | 0.067 | SG1 |
| GSR-106 | 0.875 | 0.125 | SG1 |
| GSR-124 | 0.915 | 0.085 | SG1 |
| GSR-125 | 0.968 | 0.032 | SG1 |
| GSR-132 | 0.658 | 0.342 | SG1 |
| GSR-142 | 0.954 | 0.046 | SG1 |

(*Continued*)

**Table 6.** (Continued)

| Genotypes | Inferred Ancestry | | Structure group |
|---|---|---|---|
| | Q1 | Q2 | |
| VOHP-3102 | 0.965 | 0.035 | SG1 |
| VL *Dhan* 206 | 0.878 | 0.122 | SG1 |
| VL 32197 | 0.964 | 0.036 | SG1 |
| Someshwar | 0.944 | 0.056 | SG1 |
| Bala | 0.904 | 0.096 | SG1 |
| PB-1 | 0.92 | 0.08 | SG1 |

improvement possibilities with existing resources [12, 46]. As a result of the widespread cultivation of genetically similar cultivars across a large area, the pathogen population is subjected to selection pressure, causing it to establish new races. Rice production has become a global threat as a result of the emergence of these new harmful races. However, the problem can be avoided by finding possible donors for unique functional genes or alleles that will help to overcome the disease and ensure future rice harvests [12, 37]. The present experiment investigated the genetic diversity of released varieties, advanced breeding materials, and traditional rice varieties for blast resistance genes using 25 molecular markers.

In this study, we used functional/gene-based molecular markers to genotype fifty-two rice hill germplasm collections for 25 major blast-resistant genes, in addition to field evaluations. We examined 52 rice accessions for leaf blast disease resistance in the uniform blast nursery and found that 29 (58%) and 22 (42%) genotypes were highly resistant to leaf and neck blast disease, respectively. Surprisingly, 16 accessions were found to be common for both leaf and neck blast resistance among the highly resistant rice accessions. With one released variety, VL *Dhan* 158, the vast majority of these accessions are advanced breeding materials.

Identification of the individual resistance based on phenotype is typically challenging because it is heavily influenced by developmental stage and environmental factors. However, using a linked marker associated with the *R* genes is the easiest and most reliable way for identifying individual/multiple gene(s) [47, 48]. The frequency of *R*-gene positive alleles ranged from 0% to 100%, with the genetic frequency of 25 major blast resistance genes ranging from 32% to 60%. The most positive alleles for the fifteen resistance genes are found in only two accessions (VL 8394 and VL *Dhan* 158) [49–51]. Our findings are similar to those of Yadav et al. [40] and Susan et al. [47], who reported gene frequencies ranging from 0% to 100% in 80 rice varieties released by National Rice Research Institute (NRRI), Cuttack, 9.4% to 100% in 32 Chinese rice germplasm, and 6% to 27% in 288 Indian landraces, respectively. The *R*-genes *Pib*, *Pi9*, *Pi1*, and *Pi33* appeared to be present in all rice accessions. Our findings match those of Yadav et al. [40], who discovered the *Pib* gene in all eighty rice accessions studied. Similarly, the *Pi9* gene was discovered in 51 Indian landraces [40] and 40 Chinese rice varieties [52]. However, just a few studies have documented the *Pi9* gene's rare prevalence [53, 40]. This could be owing to the *Pi9* gene's origin in the wild species *O. minuta* and its subsequent introduction into Indica rice [53]. The *Pi1* gene was detected in 39 landraces with a frequency of

**Table 7. Percentage of variation explained by the first 3 axes using blast resistance gene in PCoA.**

| Axis | 1 | 2 | 3 |
|---|---|---|---|
| Variation of the individual axis (%) | 17.18 | 12.29 | 9.21 |
| Cumulative variation (%) | 17.18 | 29.47 | 38.67 |

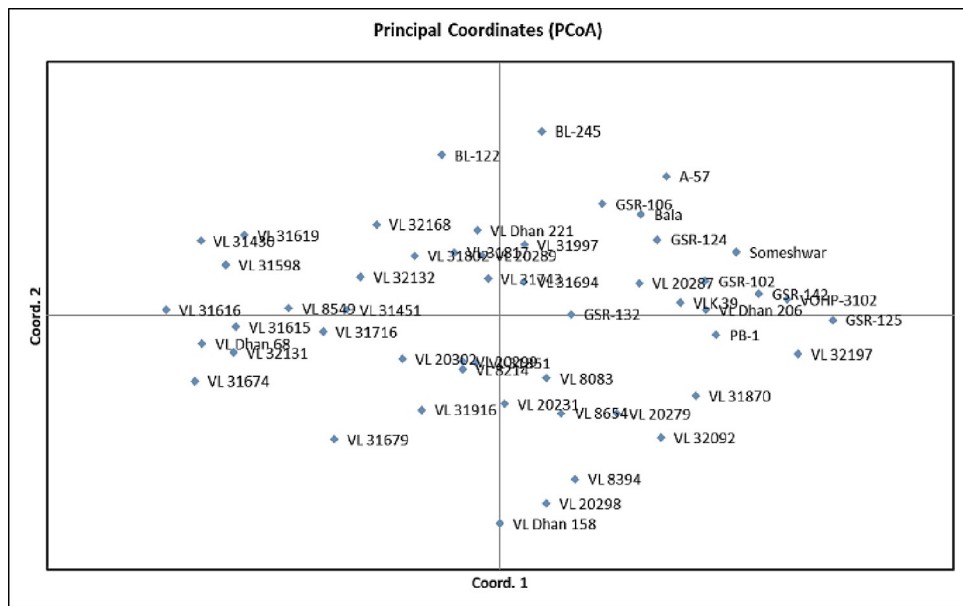

**Fig 3. PCoA of 25 molecular markers linked to blast resistance in 52 rice genotypes.**

46.98%, according to Ingole et al. [50]. The presence of the *Pi33* gene was discovered in 77 accessions in another investigation [5].

The genes *Pit*, *Pish*, and *Pikhahe-1(t)* were found in 17, 23, and 29 accessions, respectively [12, 40]. They are also found in the majority of accessions, according to earlier studies [12, 40]. In twenty-three accessions, the *Pi56* gene was found. Although it has previously been detected in 27 landraces from northeastern India [40] and 26 NRRI Cuttack, released varieties, the gene *Pi5* was found in 35 accessions [40]. Nine and twelve accessions, respectively, have the *R* genes *Piz* and *Piz-t*. However, there was no significant correlation was found between these two *R* genes and observed phenotypes. Similarly, they show partial resistance to the genotypes examined by Yadav et al. [40] and Susan et al. [47]. The *Pi2* and *Pita/Pita2* genes were detected in the majority of the rice accessions with high resistance to leaf blast [53]; however, a few genotypes without either of these genes were also resistant to leaf blast and may contain other

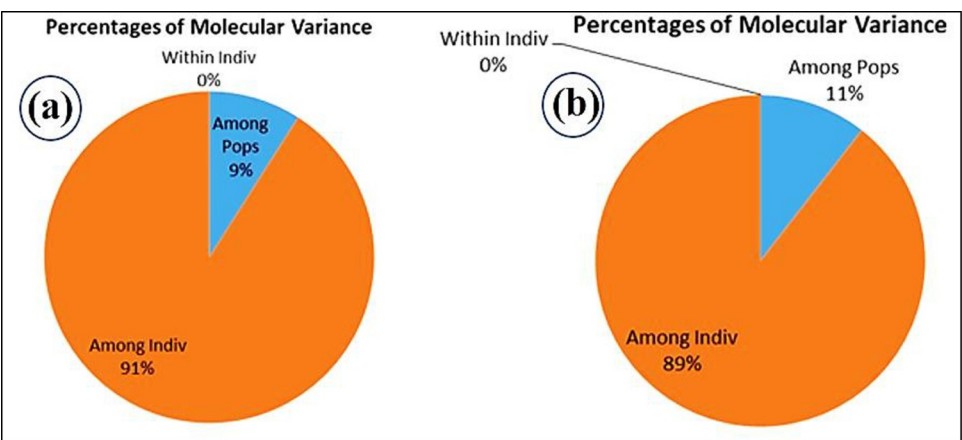

**Fig 4.** AMOVA analysis based on populations separated with leaf blast scores (a) and neck blast scores (b).

unique *R*-genes/alleles. Except for VL 20287, which tested positive for the *Pita/Pita2* gene, the genotypes that rated highly vulnerable to leaf blast did not include either of these two genes [53]. A blast resistance (BR) gene *Pik-l* was delineated in the region ~168.05 kb of the telomeric end of long of chromosome 11 of Japonica rice cv. Nipponbare. Based on its genomic position and distinct resistance spectra and also compared to previously identified *Pik* alleles, the new BR gene *Pik-l* was inferred to be a new allele of *Pik* locus [54]. Others earlier findings found that genes *Pi2* and *Pita/Pita2* express an NBS-LRR type *R* protein which are responsible to increase the resistant ability rice against leaf blast disease across a broad spectrum of pathogenic races [55,56].

In 19 accessions, the panicle blast resistance gene *Pb1* was found. Only 9 accessions were found to have high resistance to neck blast, while the other ten showed moderate resistance. The *Pb1* gene is a quantitative resistance gene that confers broad-spectrum resistance to all races. Despite having the *Pb1* gene, 10 accessions were found to have only moderate resistance to neck blast. This could be owing to the involvement of at least four QTLs in neck blast resistance, three of which, Chr7, Chr9, and Chr11, have a negative impact on *Pb1*-mediated resistance, while Chr8, on the other hand, has a positive impact. These four QTLs are expected to influence the *Pb1*-mediated resistance either individually or in combination with others [57]. As of today, a few *R* genes, *Pi25*, *Pb1*, *Pi64*, *Pi-jnw1*, and *Pi68(t)* [20–24] and QTLs like, *qNBL-9*, *qNBL-10*, *qNBL-5* [58], *qNB11-1*, *qNB11-3*, *qNB1-1*, *qNB1-2*, *qNB1-3* [59], *qPbh11-1* and *qPbh7-1*, [60] were found to confer resistance to neck blast. Among them, *Pi64*, and *Pi68(t)* were identified for the leaf as well as neck blast resistance. The *pi21* gene was discovered in just four accessions. Surprisingly, all four accessions had high resistance to neck blast, while only the two genotypes had high and moderate resistance to leaf blast disease. The *pi21* gene is a quantitative resistance gene for rice blast disease that offers broad-spectrum resistance [61].

The distance-based clustering was evaluated using genotype data, which divided the 52 germplasm into two primary groupings. Cluster I genotypes was moderately resistant to leaf and neck blast, whereas Cluster II genotypes are highly resistant to both leaf and neck blast. Similarly, the population structure analysis separated the 52 rice accessions into two subpopulations (SG1 and SG2), each with eight admixtures.

The leaf and neck blast-resistant genotypes are found in the first and second quadrants of the PCoA analysis, whereas moderately resistant genotypes were found in the first, third, and fourth quadrants. Previous research has also divided resistant and susceptible germplasm into distinct categories [40, 47]. A statistical approach for estimating molecular variance in a single species is the analysis of molecular variance (AMOVA). The AMOVA analysis revealed that there is the highest diversity within the population and minimal diversity between populations.

As a result of association mapping investigations, several genes influencing significant features have been uncovered, and it is now being utilized to deconstruct the genetic basis of many new qualities [62]. Two markers related to blast resistant genes *Pi36* and *Pik* were found to be strongly associated with neck blast resistance, whereas three markers related to blast resistant genes *Pi2*, *Pita/Pita2*, and *Pikm* were found to be significantly associated with leaf blast resistance. Previous research on association mapping and blast disease resistance has shown its effectiveness in identifying markers associated with QTLs and/or resistance genes giving blast resistance [12, 33, 40]. The identified resistant rice accessions could be used as donors in future breeding projects because they come from a variety of genetic origins. These resistant accessions might then be studied for the existence of novel functional genes/alleles, allowing them to be exploited in rice improvement programs tailored to the needs of agricultural systems.

## Conclusions

The identification of resistant germplasm for both leaf and neck blast will be facilitated by phenotyping along with the molecular characterization of blast resistance genes. Our current research on leaf and neck blast screening provided significant germplasm for breeders to employ as parent material for blast resistance transfer, particularly neck blast resistance, in the production of resistant breeding lines. Further identified resistant lines could be a valuable resource for blast resistance gene mapping, particularly in the case of neck blast disease.

## Acknowledgments

The authors are grateful to the ICAR-Vivekananda Parvatiya Krishi Anusandhan Sansthan, Almora, Uttarakhand, India and the Taif University, Taif, Saudi Arabia for providing all facilities and support during conducting the study.

## Author Contributions

**Conceptualization:** Jeevan B., Rajashekara Hosahatti, Prasanna S. Koti, Vinaykumar Hargi Devappa, Umakanta Ngangkham, Pramesh Devanna, Manoj Kumar Yadav, Krishna Kant Mishra, Jay Prakash Aditya, Palanna Kaki Boraiah.

**Data curation:** Jeevan B., Akbar Hossain.

**Formal analysis:** Jeevan B., Akbar Hossain.

**Funding acquisition:** Ahmed Gaber, Akbar Hossain.

**Investigation:** Manoj Kumar Yadav.

**Methodology:** Jeevan B., Rajashekara Hosahatti, Prasanna S. Koti, Vinaykumar Hargi Devappa, Umakanta Ngangkham, Pramesh Devanna, Manoj Kumar Yadav, Krishna Kant Mishra, Jay Prakash Aditya, Palanna Kaki Boraiah.

**Validation:** Jeevan B., Rajashekara Hosahatti, Prasanna S. Koti, Vinaykumar Hargi Devappa, Krishna Kant Mishra, Jay Prakash Aditya.

**Visualization:** Jeevan B., Rajashekara Hosahatti, Prasanna S. Koti, Vinaykumar Hargi Devappa, Krishna Kant Mishra, Jay Prakash Aditya.

**Writing – original draft:** Jeevan B., Rajashekara Hosahatti, Prasanna S. Koti, Vinaykumar Hargi Devappa, Umakanta Ngangkham, Pramesh Devanna, Manoj Kumar Yadav, Krishna Kant Mishra, Jay Prakash Aditya, Palanna Kaki Boraiah.

**Writing – review & editing:** Ahmed Gaber, Akbar Hossain.

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
