## [Decision Letter · Decision Letter 0]

21 Sep 2022

PONE-D-22-18593Phenotypic and Genotypic screening of fifty-two rice (Oryza sativa L.) germplasms for desirable cultivars against blast diseasePLOS ONE

Dear Dr.Hossain,

Thank you for submitting your manuscript to PLOS ONE. After careful consideration, we feel that it has merit but does not fully meet PLOS ONE’s publication criteria as it currently stands. Therefore, we invite you to submit a revised version of the manuscript that addresses the points raised during the review process.

We look forward to receiving your revised manuscript.

Kind regards,

Sundaram R. M., Ph.D.

Academic Editor

PLOS ONE

Journal Requirements:

    "We gratefully acknowledge the support of ICAR-Vivekananda Parvatiya Krishi Anusandhan Sansthan, Almora, Uttarakhand, India for financial support and facilities. We also thank Mr. Omkar Pratap, technical officer for his assistance in lab experiments. Also, the authors appreciated Taif University Researchers for funding this research with Supporting Project number (TURSP428 2020/39), Taif University, Taif, Saudi Arabia for providing financial support for the publication of this research.

Additional Editor Comments:

In view of the comments of the reviewers and based on my own assessment, I recommend the manuscript for a major revision

Reviewers' comments:

Reviewer's Responses to Questions

**Comments to the Author**

1. Is the manuscript technically sound, and do the data support the conclusions?

Reviewer #1: Partly

Reviewer #2: Yes

2. Has the statistical analysis been performed appropriately and rigorously? 

Reviewer #1: Yes

Reviewer #2: Yes

3. Have the authors made all data underlying the findings in their manuscript fully available?

Reviewer #1: Yes

Reviewer #2: Yes

4. Is the manuscript presented in an intelligible fashion and written in standard English?

Reviewer #1: Yes

Reviewer #2: Yes

5. Review Comments to the Author

Reviewer #1: The article is well written but I have few observations: (i) the natural screening is ok for testing the field resistance but to know the proper resistant line it is essential to go for artificial screening. Why the authors did not go for that? (ii) Neck blast is highly dependent on favorable weather condition coinciding with the flowering time. You have taken different breeding lines which must have different durations so, there is a chance that the lines which are showing resistance may be disease escape. (iii) Can we report breeding lines which we don't have IET numbers?

Reviewer #2: The paper is written in a detail way and is useful to identify the potential donors for leaf and neck blast. MAS utilization with the genes for neck and leaf blast will be helpful to develop blast resistant genotypes. The authors explained the research paper in a detail way and understandable for the researchers involved in rice breeding.

6. PLOS authors have the option to publish the peer review history of their article (what does this mean?). If published, this will include your full peer review and any attached files.

Reviewer #1: No

Reviewer #2: **Yes: **Dr. S V Sai Prasad

---

## [Author Response · Author response to Decision Letter 0]

30 Sep 2022

Response to reviewers' comments

We are thankful to both the reviewers and editor for their positive feedback and valuable suggestions to improve our manuscript. 

We have addressed all the queries raised by the reviewers. Our responses to the reviewer’s queries are listed below. 

Reviewer #1

1. The natural screening is ok for testing the field resistance but to know the proper resistant line it is essential to go for artificial screening. Why the authors did not go for that? 

Author's response: We agree with the reviewer's observation. We used two isolates of M. oryzae to conduct artificial screening for leaf blast against entries that demonstrated high resistance in the field conditions and found resistance even at greenhouse experiments in order to confirm the field results. However, till date, full proof artificial screening method for the neck blast disease is not available (therefore, field screening under high disease pressure is more reliable technique so far). The susceptible check in our study recorded the highest disease score, indicating the high disease pressure in the field and the result obtained doesn’t require any further confirmation through artificial screening. Moreover, the field evaluations were conducted at Almora, India, which has been designated as a hotspot location for the rice blast disease. Therefore, the data obtained in the filed screening is reliable and consistent over three years of repeated screening. 

2. Neck blast is highly dependent on favorable weather condition coinciding with the flowering time. You have taken different breeding lines which must have different durations so, there is a chance that the lines which are showing resistance may be disease escape. 

Author's response: Reviewer raised a valid question. Except for a few lines that flowered in 105–107 days, the majority of the rice lines used in this research flowered between 95 and 100 days. The days to flowering were therefore not significantly different across the lines employed in this investigation. In all three years of the experiment (2018 to 2020), the sowing was carried out during the first week of June. Over the course of the study period, we observed maximum mean temperatures of 31 ⁰C and minimum mean temperatures of 19 ⁰C, with an average mean temperature of ~23 ⁰C and a mean relative humidity of >75%, which is extremely favourable for the rice blast disease. Therefore, no question of disease escapes due to weather parameters. 

3. Can we report breeding lines which we don't have IET numbers?

Author's response: Yes, we are willing to report the breeding lines without an IET number. Since entries participating in AICRP trials are the only ones to which IET numbers can be assigned. Therefore, no issue with reporting without IET numbers. 

Additional Comments from the reviewer:

1. L.42 - In order to explore the rice blast-resistant sources, we initially performed a large-scale screening of 277 rice accessions → In order to explore the rice blast-resistant sources, initially performed a large-scale screening of 277 rice accessions.

Authors: Agreed and included the suggestion

2. L.328 to 331 - Susan et al. [52] examined 288 landraces for rice blast disease resistance and discovered that 75 were highly resistant, 127 were moderately resistant, and 86 were found susceptible. Another study looked at 358 rice accessions for resistance to neck blast and found that 124 cultivars were resistant and 234 cultivars were susceptible, respectively [52]. → delete

Authors: Agreed and deleted as per suggestions.

3. L.335 to 336 - The identification of blast R genes in various germplasms can be done with the use of linked molecular markers [49,50]. → delete

Authors: Agreed and deleted as per suggestions.

4. L.381 to 383 - Surprisingly, the population structure may be able to distinguish between resistant, moderately resistant, and susceptible germplasm. Similarly, the population structure was able to differentiate the 80 NRVs and 288 germplasm into resistant and susceptible [12,47]. → delete

Authors: Agreed and deleted as per suggestions.

Reviewer #2

Authors’ response: Reviewer-2 comments in the PDF file have been incorporated in the text of the manuscript. Please check all edits in track change mode.

---

## [Decision Letter · Decision Letter 1]

8 Jan 2023

Phenotypic and Genotypic screening of fifty-two rice (Oryza sativa L.) germplasms for desirable cultivars against blast disease

PONE-D-22-18593R1

Dear Dr. Hossain,

We’re pleased to inform you that your manuscript has been judged scientifically suitable for publication and will be formally accepted for publication once it meets all outstanding technical requirements.

Kind regards,

Muhammad Abdul Rehman Rashid, PhD

Academic Editor

PLOS ONE

Additional Editor Comments (optional):

Reviewers' comments:

Reviewer's Responses to Questions

**Comments to the Author**

1. If the authors have adequately addressed your comments raised in a previous round of review and you feel that this manuscript is now acceptable for publication, you may indicate that here to bypass the “Comments to the Author” section, enter your conflict of interest statement in the “Confidential to Editor” section, and submit your "Accept" recommendation.

Reviewer #2: All comments have been addressed

2. Is the manuscript technically sound, and do the data support the conclusions?

Reviewer #2: Yes

3. Has the statistical analysis been performed appropriately and rigorously? 

Reviewer #2: Yes

4. Have the authors made all data underlying the findings in their manuscript fully available?

Reviewer #2: Yes

5. Is the manuscript presented in an intelligible fashion and written in standard English?

Reviewer #2: Yes

6. Review Comments to the Author

Reviewer #2: The author has incorporated all the minor suggestions made and able to answer the issues raised by both the reviewers. The information generated properly and was written in a detail way and is useful to identify the potential donors for leaf and neck blast. MAS utilization with the genes for neck and leaf blast will be helpful to develop blast resistant genotypes. The authors explained the research paper in a detail way and understandable for the researchers involved in rice breeding.

7. PLOS authors have the option to publish the peer review history of their article (what does this mean?). If published, this will include your full peer review and any attached files.

Reviewer #2: No

---

## [Editor Report · Acceptance letter]

26 Jan 2023

PONE-D-22-18593R1 

Phenotypic and Genotypic screening of fifty-two rice (*Oryza sativa* L.) genotypes for desirable cultivars against blast disease 

Dear Dr. Hossain:

I'm pleased to inform you that your manuscript has been deemed suitable for publication in PLOS ONE. Congratulations! Your manuscript is now with our production department. 

Kind regards, 

on behalf of

Dr. Muhammad Abdul Rehman Rashid 

Academic Editor

PLOS ONE